# Study on the Relationship between Worker States and Unsafe Behaviours in Coal Mine Accidents Based on a Bayesian Networks Model

**Zhaobo Chen [1,\*], Gangzhu Qiao [2] and Jianchao Zeng [2]**

[1] Key Research Bases for Humanities and Social Sciences in Shanxi: Research Center for Innovation and Development of Equipment Manufacturing Industry, Taiyuan University of Science and Technology, Taiyuan 030024, China

[2] Division of Big Data and Visual Computing, North University of China, Taiyuan 030051, China

\* Correspondence: chenzb1983@tyust.edu.cn; Tel.: +86-351-277-6604

**Abstract:** Unsafe behaviours, such as violations of rules and procedures, are commonly identified as important causal factors in coal mine accidents. Meanwhile, a recurring conclusion of accident investigations is that worker states, such as mental fatigue, illness, physiological fatigue, etc., are important contributory factors to unsafe behaviour. In this article, we seek to provide a quantitative analysis on the relationship between the worker state and unsafe behaviours in coal mine accidents, based on a case study drawn from Chinese practice. Using Bayesian networks (BN), a graphical structure of the network was designed with the help of three experts from a coal mine safety bureau. In particular, we propose a verbal versus numerical fuzzy probability assessment method to elicit the conditional probability of the Bayesian network. The junction tree algorithm is further employed to accomplish this analysis. According to the BN established by expert knowledge, the results show that when the worker is in a poor state, the most vulnerable unsafe behaviour is violation, followed by decision-making error. Furthermore, insufficient experience may be the most significant contributory factor to unsafe behaviour, and poor fitness for duty may be the principal state that causes unsafe behaviours.

**Keywords:** coal mine accident; worker state; unsafe behaviour; Bayesian networks

---

## 1. Introduction

Despite efforts, at different levels, to achieve safety in coal mines, such as the innovative use of new technology and the implementation of safety-related regulations, the occurrence of accidents and incidents is a concern. Mining is the highest-risk industry in the world: its rate of occurrence of accidents is up to 10 times that of other industries [1]. Hence, many researchers have been drawn to analyse the underlying mechanisms of coal mine accidents to reduce their frequency.

Human error has been demonstrated as the primary factor in mine accidents. The US Bureau of Mines concludes that nearly 85% of all mining accidents were attributed to human errors as the causal factors. In China, the principal causes of coal mine accidents are human factors or human error, accounting for 95.10% [2] of all accidents. Consequently, there have been numerous studies concerning the causes of accidents from the perspective of human factors in a mining context. For instance, Paul and Maiti [3] examined the role of behavioural factors in the occurrence of mine accidents and injuries through a case study, and concluded that negative affectivity and job dissatisfaction caused workers to take more risks and behave unsafely. Casey and Krauss [4] investigated the relationships between the error management climate and miner safety performance, and found that co-worker safety support and safety communication exhibited particularly strong relationships with safety

performance. Dahl [5] identified the most significant factors within the organizational context, which affects the worker's knowledge of rules and procedures on the basis of semi-structured interviews; in particular, these factors can be sorted into three paramount categories: the safety management system, work characteristics and social interaction. Many accident investigations also provide the following information about potential reasons for coal mine accidents, for example:

> "Lacking the necessary safety awareness, leaders of the ventilation team and tile inspection team fail to make a timely arrangement for the hidden danger after they receive a notification, neither do they implement any monitor or inspection in this area."

> (Notification of gas gauge accident in Hexi Coal Mine at 13:24 on 10 November 2010)

> "Because of a lack of necessary skill, the operator in the power substation executed the power dispatching command inaccurately and did not implement the instruction of switch 35 kV bus tie 370 from "Operation" to "Highly available", and then caused an accidental power cut."

> (Notification of power cut accident in Hedong Coal Mine at 18:01 on 2 September 2012)

Obviously, a recurring conclusion of accident investigations is that a poor worker state, such as lacking necessary safety awareness and skill in the above two accidents, is an important contributory factor to unsafe behaviour. Recently, researchers also have shown that the state of workers directly leads to unsafe acts causing serious accidents. For instance, early morning awakening and non-restorative sleep are significantly associated with an increased likelihood of minor non-fatal accidents during work and leisure time [6]. Health issues, such as epilepsy, diabetes and other long-term diseases, are major factors affecting driving ability and road safety [7]. Experience can also be counted as a significant human factor that affects individual human performance [8]. In this article, we seek to quantify the impact of worker states on unsafe behaviour in coal mine accidents based on a case study in China.

To date, there have been numerous projects studying human error in accidents, such as those by [9–11] and [12]. At present, the Human Factors Analysis and Classification System (HFACS) is a human error analysis method that is able to assist investigators in the identification of human and organisational factors [13]. Briefly, HFACS categorizes human errors of operators, including decision errors, skill-based errors, perceptual errors, and violations, combined with latent conditions upstream in the organisation. HFACS is now widely used to investigate and analyse human factors involved in accidents. For example, a sample of mining incidents in Australia was analysed using HFACS [14]. A modified version of HFACS was used to analyse accident cases from across the state of Queensland to identify human factor trends within mining [15]. Liu [16] established a human factor analysis and classification system for China's mines (HFACS-CM) based on the statistical results of 362 major coal mine accidents in China, and investigated the poor safety practices of coal miners and their related influencing factors. Most of these works mainly use statistical methods to examine the relationship between human errors in an accident.

As an uncertain processing model for simulating the causality in human reasoning, Bayesian networks (BN), also known as Bayesian Belief Networks (BBN), were first introduced in the late 1980s by [17]. Since then, an increasing number of successful applications of such networks to different problem domains have been developed, which demonstrates that they have established their position in Artificial Intelligence as valuable representations of reasoning with uncertainty. The network has, in recent years, been implemented in diverse research areas related to safety. For instance, Wang [18] examined human reliability through a BN model and human factors experiments. Garcia-Herrero [19] used BN to determine the relationships between organisational culture and safety culture in a nuclear power plant. Li [20] developed a fuzzy BN approach to improve the quantification of organisational influences in human reliability analysis frameworks. Zhou [21] proposed a quantitative human reliability analysis model based on fuzzy logic theory, BN, and a cognitive reliability and error analysis method for the tanker shipping industry. The BN modelling process is presented in Figure 1.

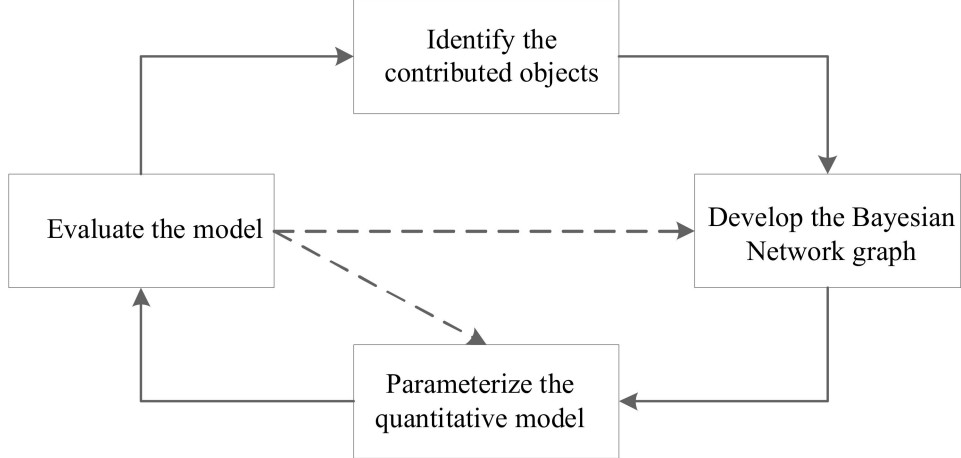

**Figure 1.** The cycle to build the Bayesian networks [22].

This work contributes to the literature by integrating two subjects of current interest: human error analysis and Bayesian networks, into one model. However, investigations of coal mine accidents in China mainly focus on the cognizance of the responsibility for the accident, and the worker state (mental fatigue, anoxia, illness, medical abnormalities, physiological fatigue, etc.) is rarely reflected. Consequently, we are unable to compose an up-to-date, large, rich dataset from investigations of Chinese mine accidents to propose a Bayesian network covering the influence of worker states on unsafe behaviour. Hereby, we take two steps to establish the network: first, according to the HFACS four-level structure, the graphical structure of the Bayesian network is constructed, where states of the worker include: adverse mental states, adverse physiological states, physical/mental limitations, and fitness for duty, while the unsafe acts include: decision errors, skill-based errors, perceptual errors, and violations. We use these phrases to identify the variables in the network, and their definitions may be found in [13]. Second, with the help of three experts from a coal mine safety bureau, we propose a verbal versus numerical fuzzy probability assessment method to elicit the conditional probabilities of the network to handle the problem of insufficient data. In particular, the junction tree algorithm is used to model inference in this work.

The rest of the paper is organised as follows: Section 2 identifies the factors involved in worker states and builds the structure of the BN. Section 3 proposes a verbal versus numerical fuzzy probability assessment method to elicit the conditional probability of the BN to quantify the influence of different factors on unsafe behaviours, and the proposed BN is evaluated in Section 4. In Section 5, the junction tree algorithm is used to accomplish the inference, Section 6 analyses the results, and conclusions are drawn in Section 7.

For convenience, we will use the phrase "behaviour network" to refer to our networks for the worker state and unsafe behaviour herein.

## 2. Structure of the Behaviour Network

Actually, BNs include a qualitative and a quantitative part: the qualitative part is a graphical structure, where the nodes represent the variables and the arcs represent their dependencies. The quantitative part is indicated by conditional probabilities, which represent the strengths of the relationships between these variables.

According to the HFACS four-level structure, we identify the four variables: unsafe worker behaviours include: Decision Errors (DE), Skill-based Errors (SBE), Perceptual Errors (PE), and Violations (V), as worker's unsafe behaviours. Simultaneously, the states of the worker, Adverse Mental States (AMS), Adverse Physiological States (APS), Physical/Mental Limitations (PML), and Fitness for Duty (FD), can be furtherly reflected by other more-concrete states. First, based on 163 accident investigation reports from the Fenxi Coal Mine Safety Bureau in Shanxi (the largest coal-producing province in

China), we adopt six direct and indirect worker states related to unsafe behaviour, e.g., Inadequate Safety Awareness (ISA), Poor Vigilance Awareness (PVA), Insufficient Experience (IE), Insufficient Competencies (IC), Poor Situation Awareness (PSA), and Alcoholic Intoxication (AI), since most of them are proven as being the main individual factors that affect unsafe behaviors in [23]. Actually, the above six worker states are directly refined from accident investigation reports, and some example are listed in Table 1.

**Table 1.** The worker states in coal mine accidents.

| Name of Accident | Causes of Accident * | Refined Worker States |
|---|---|---|
| Landslide accident in Hexi Coal Mine on 23 January 2010 | 1. Workers' safety awareness was insufficient. 2. The worker was impercipient about the potential hazards. 3. Conclusions of the geological survey report were erroneous given the inadequate skill of the geological exploration. | Inadequate Safety Awareness Poor Vigilance Awareness Insufficient Competencies |
| Blackout accident of main fan in Zhenghang Coal Mine on 21 September 2007 | 1. The skill of the main fan driver was poor. 2. Short circuit was caused by the overhaul of the power supply team, and the potential danger was not found. | Insufficient Competencies Poor Vigilance Awareness |
| Gas excess accident in 1511 cutting roadway of Longyang Coal Mine on 26 December 2010 | 1. The pillar strength was insufficient because of the insufficient experience of workers. 2.The problem was found by a security officer but not paid attention to, and the gas prevention consciousness was insufficient. | Insufficient Experience Inadequate Safety Awareness Poor Vigilance Awareness |
| Gas asphyxiation accident of Songjiatang Coal Mine on 18 May 2007 | 1. After drinking, the workers went to the mine and unlocked the fence to enter the blind lane. 2. The gas inspector had no job qualification certificate. | Alcoholic Intoxication Insufficient Competencies |

\* "Causes of accident" are directly quoted from the investigation reports of the coal mine accident.

Furthermore, we identify another four worker states from relevant literatures:

(1) Mental Fatigue (MF) and Physiological Fatigue (PF). From a psychological perspective, mental fatigue results from prolonged periods of cognitive activity and leads to a decline in the cognitive and behavioral performance [24,25]. Roske-Hofstrand [26] also observed that 21% of the reported incidents in the Aviation Safety Reporting System mentioned factors related to fatigue for both pilots and air traffic controllers. Consequently, we conclude that the two kinds of fatigue can lead to unsafe behaviour.

(2) Illness (I). Rolison [27] showed that illness increased the number of road accidents; for example, 84% patients helped cause automobile accidents after an epileptic seizure [28]. Hereby, we adopt illness as a worker state related to unsafe behaviour.

(3) Medication Effects (ME). Khoshakhlagh [29] pointed that the consumption of Gemfibrozil and Glibenclamide can lead to more traffic accidents.

Finally, based on a survey of three experts from the Fenxi Coal Mine Safety Bureau, we identify another two variables for the behaviour network: Physical Limitations (PL), and Mental Limitations (ML). The three experts are coal mine accident investigators, with an average working time of 8 years. In particular, one of the experts is a member of the technical investigation team, and he is mainly responsible for the accident scene investigation, the identification of accident elements, and the identification of the nature and direct causes of the accident. Another two experts are from the management investigation team, and they are mainly responsible for identifying the indirect causes of the accident and determining the responsibilities of relevant personnel. The three experts unanimously agreed that Physical Limitations (PL), and Mental Limitations (ML) would be related to unsafe behaviours. The conclusion was also verified by some studies showing that diabetes and vision weakness, for example, may also lead to more accidents [29]. Additionally, since the education level

of a coal mine worker is relatively low, some workers lack a general safety knowledge, and safety training seems to be ineffective for them. Consequently, we refined this as Mental Limitations (ML). The detailed description of each variable is provided in Table 2.

**Table 2.** The BN variables.

| Variables | Description |
|---|---|
| Unsafe behaviours | |
| Skill-based Errors (SBE) | Inadvertent activation/deactivation of switches, forgotten intentions, omitted items in checklists. |
| Decision Errors (DE) | Poorly executed procedures, improper choices, misinterpretation or misuse of information. |
| Perceptual Errors (PE) | Misinterpretation of the environment information, such as distances or altitude, and responding incorrectly to a variety of visual illusions. |
| Violations (V) | Violation of the relevant rules and regulations of enterprise, or relevant provisions of the state. |
| Worker states | |
| Inadequate Safety Awareness (ISA) | Poor safety awareness that may be caused by inadequacy of safety training or personality of employee. |
| Mental Fatigue (MF) | Psychological tiredness that can be caused by reasons such as pressure and boring nature of the work. |
| Poor Vigilance Awareness (PVA) | Cannot detect the possible occurrence of accidents or signs of danger in time, which may be caused by insensitivity or neglect of some potential hazards. |
| Poor Situation Awareness (PSA) | Failure to respond quickly to unexpected events, or cannot use knowledge and experience to deal with the situation effectively. |
| Medication Effect (ME) | Taking drugs with side effects on psychological or mental conditions, such as cold medicine or analgesics. |
| Illness (I) | Common diseases such as a cold or fever. |
| Alcoholic Intoxication (AI) | Excessive drinking that affects worker's physiological state and attention, judgment, etc. |
| Physiological Fatigue (PF) | Physical exhaustion that may be caused by poor diet, lack of sleep, or lack of rest. |
| Insufficient Experience (IE) | Insufficiency of experience needed to deal with the situation effectively. |
| Insufficient Competencies (IC) | Inadequate duty-required knowledge and skills. |
| Physical Limitations (PL) | Physical limitations that may adversely impact performance such as poor vision, lack of physical strength, some chronic diseases, or problems related to speech, language or hearing. |
| Mental Limitations (ML) | Mental limitations that affect performance such as lack of general knowledge or poor education. |

The value sets associated with the above variables are 0 and 1, where 0 indicates that the corresponding factor does not occur, and 1 indicates that it does. In particular, the worker states listed in Table 1 can be divided into the four basic categories defined in HFACS. Mental fatigue, poor vigilance awareness, poor situation awareness, and inadequate safety awareness are adverse mental states (AMS). Illness and physiological fatigue are adverse physiological states (APS). Insufficient competencies, insufficient experience, medical effect, and alcoholic intoxication relate to the fitness for duty (FD). Physical limitations and mental limitations are physical/mental limitations (PML) of workers. Consequently, we introduce the above four variables as auxiliary variables to simplify the conditional dependency structure. Furthermore, according to the description and the definition of the variables, we propose the following two assumptions:

**Assumption 1.** *Unsafe behaviours (SBE, DE, PE, and V) are the leaf nodes in the behaviour network and are independent of each other.*

**Assumption 2.** *States (I, PL, ML, AI, IE, ME, and ISA) are root nodes in the behaviour network and are independent of each other.*

After the nodes and their values are formed as sets, we will build the BN structure of our behaviour network. As the graphical structure of a BN represents the causal relationships between the variables, to the experts this kind of relationship is not difficult to identify. However, the knowledge of experts is scattered in their minds; we will take a certain method to elicit this knowledge, in the form of question and answer sessions. For instance, does excessive drinking increase the probability of "poor vigilance awareness"? If the answer is "yes", then a directed edge is inserted between the two nodes from "Alcoholic Intoxication (AI)" to "Poor Vigilance Awareness (PVA)". However, by being required to pay special attention to the relationships between these nodes, including the direct and indirect causal effects, we only need direct effects to construct the network structure. Experience shows that experts find it difficult to distinguish the direct from the indirect effects, making it easy to elicit an incorrect Bayesian structure. Hence, we checked the structure to delete indirect relationships after experts proposed the relationship between the domain variables involved in the behaviour network. With the help of three experts from the Fenxi Coal Mine Safety Bureau, the relationships between these variables are given in Appendix A (Enclosed Table A1). Based on the relationships between the variables in this behaviour network, the Bayesian structure of the behaviour network is given in Figure 2.

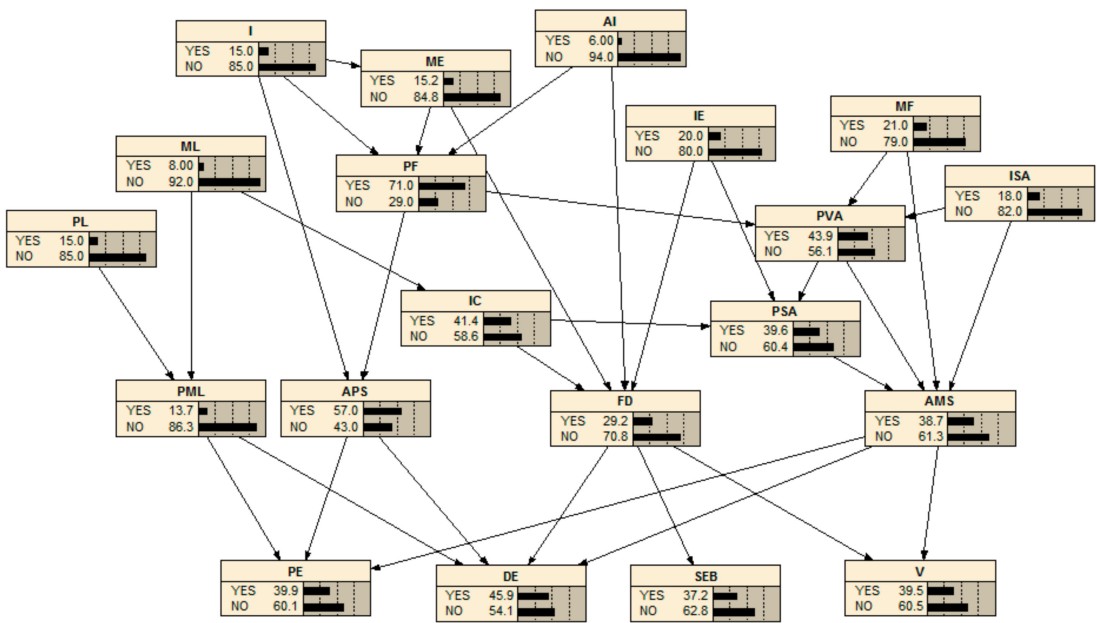

**Figure 2.** The structure of the behaviour network.

## 3. Elicitation of BN Parameters for the Behaviour Network

The BN parameters are the probability distribution of node *X* conditional upon *X*'s parents, representing the strengths of the relationships between the variables, which are the kernel of the behaviour network [30]. Based on the conditional independence resulting from the d-separation concept, the BN parameters represent the joint probability distribution $P(X)$ of the variables $X = \{X_1, X_2, \ldots, X_n\}$ as:

$$P(X) = \Pi_{i=1}^n P(X_i|pa(X_i)) \tag{1}$$

where $pa(X_i)$ is the parent set of $X_i$ for any $i = 1, 2, \ldots, n$.

Notwithstanding this, there have been numerous methods, such as the maximum likelihood approach [31], Bayesian estimation [32] and EM (expectation-maximisation) algorithms [33], used to estimate the BN parameters: this task is generally acknowledged to be daunting. In particular, worker states, such as mental fatigue, illnesses, medical abnormalities, physiological fatigue, etc., are rarely reflected in mining accident investigation reports in China. Consequently, in the event

that a large data set is unavailable, the BN parameters must be elicited from domain experts. Fortunately, a number of methods have been incorporated into the field of decision analysis for the elicitation of probabilities in such problems, and these include: the verbal versus numerical probability assessment [34], the probability scale [35] and the gamble method [36]. The probability scale and the gamble method have been proven to be effective ways of eliciting probabilities and are widely used in the construction of decision-analytic models. Unfortunately, we encountered numerous problems when using these two methods with our experts to assess the probabilities required for the behaviour network. Most importantly, it is well known that using the two methods tends to take considerable time with every single assessment. Hence, here we used the method of verbal versus numerical probability assessment to elicit the conditional probabilities of the behaviour network.

Assessing probabilities in words (e.g., somewhat possible) and numbers (e.g., 30%), are two methods with apparent differences [37]. Numerical values are more objective and accurate than verbal assignments. However, words have additional non-numeric semantic properties, and verbal assessments are easier and more natural for individuals [38]. Luckily, at present, a large body of literature finds surprisingly few differences between verbal and numerical probability assessments (e.g., [34] and [39]). There are many methods that can be applied to elicit a verbal versus numerical probability assessment. Renooij and Witteman [40] combined them to yield a novel, efficient method with which experts provided the required probabilities at a rate of over 150 per hour. In particular, the method includes a double scale with verbal as well as numerical anchors and a presentation format for the probabilities. The details of this method can be found in [34] and [40]. Since verbal probabilities are vaguer than numerical values, we will integrate this method with a fuzzy set of numerical probabilities to assess the parameters of our behaviour network. To facilitate the analysis, triangular fuzzy numbers and linguistic variables are used to access the conditional probabilities of the behaviour network here. The linguistic variables, including certain, probable, expected, fifty-fifty, uncertain, improbable, and impossible, are shown in Figure 3, along with the membership functions.

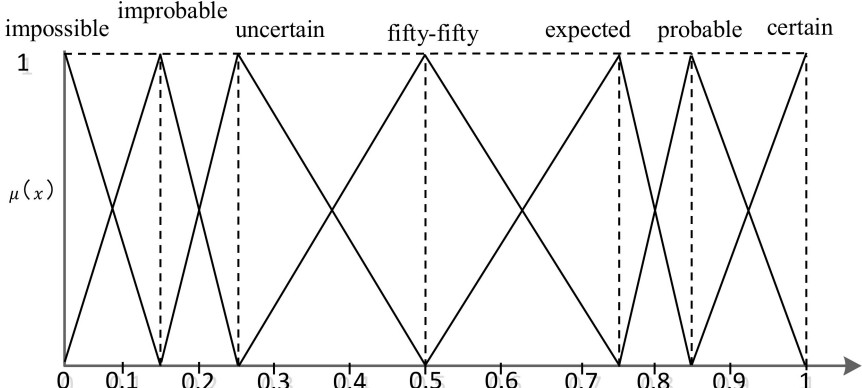

**Figure 3.** The membership function for linguistic variables.

The questionnaire for the verbal and numerical probability assessment, for instance the probability that a worker's physiological states is poor (PF = 1) given that the worker is sick and taking some medication (I = 1, ME = 1, AI = 0), is presented in Figure 4.

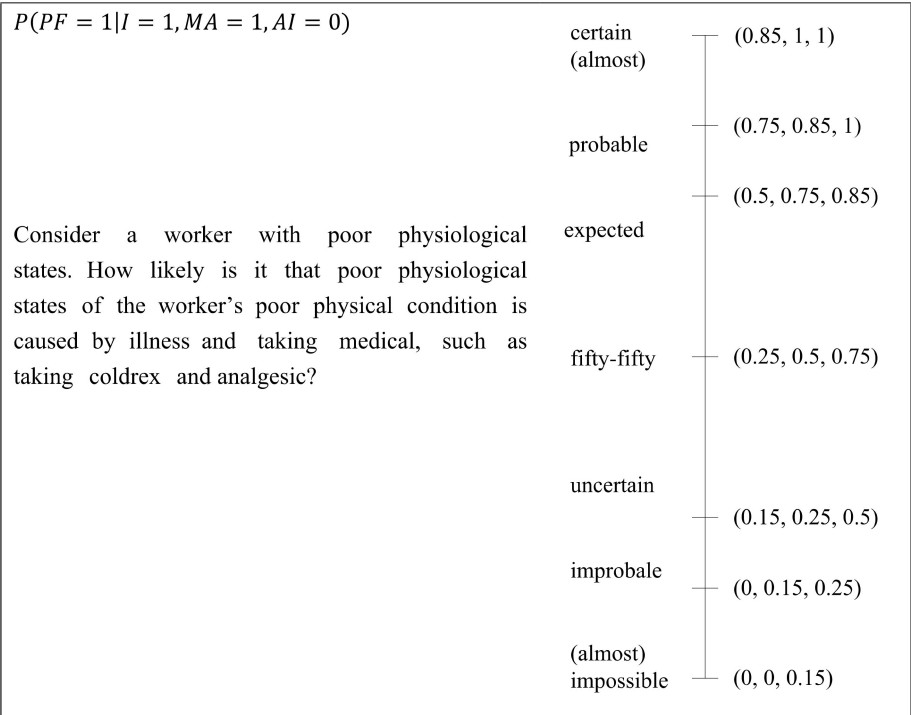

**Figure 4.** The conditional probability distribution for adverse physiological states, given an illness and taking medication.

Consequently, expert opinion can be transformed into a triangular fuzzy probability. Furthermore, if there are more than one expert, the following method can be used to aggregate their opinions.

Suppose there are $g$ experts. Let $\widetilde{P}_{ij}^k = \left(a_{ij}^k, m_{ij}^k, b_{ij}^k\right)\$$ denote the fuzzy probability of variable $i$ (the value of variable $i$ is $j$, $j = 0$, 1), which is assessed by expert $k$ ($k = 1, 2, \ldots g$). Furthermore, we use the arithmetic average method to get a more reasonable fuzzy probability $\widetilde{P}'_{ij}$ for our network:

$$\widetilde{P}'_{ij} = \omega_1 \widetilde{P}_{ij}^1 \bigoplus \omega_2 \widetilde{P}_{ij}^2 \bigoplus \ldots \bigoplus \omega_g \widetilde{P}_{ij}^g = \left(a'_{ij}, m'_{ij}, b'_{ij}\right) \tag{2}$$

where $\omega_{k_1} \widetilde{P}_{ij}^{k_1} \bigoplus \omega_{k_2} \widetilde{P}_{ij}^{k_2} = \left(\omega_{k_1} a_{ij}^{k_1} + \omega_{k_2} a_{ij}^{k_2}, \ \omega_{k_1} m_{ij}^{k_1} + \omega_{k_2} a_{ij}^{k_2}, \ \omega_{k_1} b_{ij}^{k_1} + \omega_{k_2} a_{ij}^{k_2}\right)$, $\omega_k$ is expert $k$'s weight, which can evaluate the reliability of that expert: $\omega_k$ is a comprehensive representation of the expert's knowledge, experience, and ability, and can be determined in many ways (e.g., a principal component analysis). The fuzzy probability $\widetilde{P}'_{ij}$ will be defuzzified into a crisp value as follows [41]:

$$\widetilde{P}_{ij} = \frac{a'_{ij} + 2m'_{ij} + b'_{ij}}{4} \tag{3}$$

We use this method to elicit the probabilities from three domain experts in the construction of the quantitative part of the behaviour network. Some of the parameters are given in Appendix A (Enclosed in Table A2).

## 4. Evaluation of the Behaviour Network

In the two previous sections, we proposed a behaviour network based on the collected knowledge of domain experts. In this section, we will investigate whether the established behaviour of the BN is aligned with this expert intuition. Generally, the investigation includes two parts: the testing of the dependence relationships between the variables and the judgement of the model parameters. Although the dependence relationships between the variables can be tested using the definition

of conditional independence, the mutual information between variables is in practice widely used. The mutual information $I(X, Y)$ is used to measure the effect of variable $X$ on variable $Y$, where $I(X, Y)$ is defined as:

$$I(X, Y) = -H(X) - H(X|Y) \tag{4}$$

where $H(X)$ is the entropy of a distribution over variable $X$, $H(X|Y)$ is the conditional entropy of $X$ and $Y$, and:

$$H(X) = -\sum_{x \in X} P(x) \log P(x)$$

$$H(X|Y) = -\sum_{x \in X, \, y \in Y} P(x, y) \log P(x|y) \tag{5}$$

The mutual information is non-negative $I(X, Y) \geq 0$ with equality if and only if $X$ and $Y$ are independent. Furthermore, the errors in the network parameters can be investigated by altering each parameter for the query nodes and observing the related changes in the posterior probabilities of a given query (i.e., a sensitivity analysis technique). This analysis investigates the effect of the probability parameter changes of observable nodes on the query variable and allows the expert to identify whether or not a variable is sensitive to other variables in a particular context; this can help to identify errors in either the network structure or the parameters. As the number of variables in the behaviour network is considerable, it is difficult to manually perform a sensitivity analysis; software such as Netica can be used instead. As an example, the mutual information and effect of query variable $V$ on the other variables is shown in Table 3.

**Table 3.** The mutual information and effect of query variable V on the other variables.

| Node | No Evidence | | MF = 1 | | AI = 1, MF = 1 | |
|---|---|---|---|---|---|---|
| | **Mutual Info** | **Variance of Belief** | **Mutual Info** | **Variance of Beliefs** | **Mutual Info** | **Variance of Beliefs** |
| V | 0.96825 | 0.2390766 | 0.99181 | 0.2471684 | 0.96902 | 0.2393407 |
| AMS | 0.18263 | 0.0591412 | 0.15263 | 0.0503676 | 0.12298 | 0.0407551 |
| FD | 0.08587 | 0.0286550 | 0.04775 | 0.0157284 | 0.05782 | 0.0188174 |
| DE | 0.03859 | 0.0127221 | 0.02416 | 0.0082595 | 0.02080 | 0.0069186 |
| PSA | 0.03589 | 0.0119248 | 0.01033 | 0.0035258 | 0.00834 | 0.0027553 |
| PVA | 0.03136 | 0.0103779 | 0.00647 | 0.0022256 | 0.00576 | 0.0019292 |
| SBE | 0.02058 | 0.0068608 | 0.01108 | 0.0037658 | 0.01547 | 0.0050991 |
| MF | 0.01958 | 0.0066131 | — | — | — | — |
| IC | 0.01808 | 0.0060088 | 0.00836 | 0.0028512 | 0.00613 | 0.0020214 |
| IE | 0.01527 | 0.0051583 | 0.00705 | 0.0023783 | 0.00538 | 0.0017440 |
| ISA | 0.01232 | 0.0041647 | 0.00459 | 0.0015500 | 0.00388 | 0.0012606 |
| PE | 0.00448 | 0.0014884 | 0.00294 | 0.0010049 | 0.00256 | 0.0008462 |
| ME | 0.00260 | 0.0008740 | 0.00152 | 0.0005160 | 0.00076 | 0.0002500 |
| I | 0.00117 | 0.0003923 | 0.00068 | 0.0002328 | 0.00040 | 0.0001315 |
| PF | 0.00114 | 0.0003764 | 0.00031 | 0.0001070 | 0.00057 | 0.0001913 |
| AI | 0.00082 | 0.0002766 | 0.00047 | 0.0001598 | — | — |
| APS | 0.00054 | 0.0001782 | 0.00017 | 0.0000572 | 0.00026 | 0.0000875 |
| ML | 0.00054 | 0.0001782 | 0.00023 | 0.0000778 | 0.00017 | 0.0000552 |
| PML | 0.00005 | 0.0000174 | 0.00002 | 0.0000082 | 0.00002 | 0.0000058 |
| PL | 0.00000 | 0.0000000 | 0.00000 | 0.0000000 | 0.00000 | 0.0000000 |

Based on the mutual information and sensitivity analysis, we can adjust both the structure and parameters of the behaviour network until the experts are satisfied with the system response to the test queries.

## 5. Bayesian Inference of the Behaviour Network

In this section, we update the probability estimate for a hypothesis as additional evidence is being acquired. Sometimes, this process is termed "Bayesian inference". Briefly, in the philosophy of decision theory, Bayesian inference derives the posterior probability as a consequence of two antecedents,

a prior probability and a "likelihood function" derived from a probability model for the data to be observed. Given new information, Bayesian inference uses Bayes' theorem to update the prior occurrence probability of objects. The new information, called evidence *E*, is usually obtained during the system operation. Equation (6) is used for the probability prediction in a given Bayesian network:

$$P(H|E) = \frac{P(E|H)P(H)}{P(E)} \tag{6}$$

Bayesian inference methods can be divided roughly into exact inference methods and approximate inference methods, where the exact inference methods are sometimes NP-hard (Non-deterministic Polynomial-time hard) problems. Consequently, the junction tree algorithm is used to find the inferences here. The algorithm can be summarised as follows:

**Step 1:** Moralise. Transformation of the directed graph into an undirected graph. The moralisation step entails adding edges between parents of nodes and dropping the directions of the directed graph.

**Step 2:** Triangulating the graph. Adding edges to a triangulated graph so that every cycle does not exceed three nodes.

**Step 3:** Forming the junction tree. Identifying the maximal cliques from the triangulated graph. A clique can be regarded as a node in the clique graph. If two adjacent cliques intersect with same common nodes, then the two cliques are joined by an edge labelled with shared nodes.

**Step 4:** Assigning potentials and initialising. Setting the clique potentials to the original potentials over the undirected graph.

**Step 5:** Selecting an arbitrary root node.

**Step 6:** Carrying out message passing. Selecting the nodes that are connected to only one neighbour and using standard message passing algorithms to pass messages to the graphical modes.

**Step 7:** Evaluating desired marginal potentials. Incorporating evidence, and then reading off the clique marginal potentials from the junction tree.

## 6. Results and Discussion

In the section, we will use the established behaviour network to analyse the impacts of the worker state on unsafe behaviour by inference. First, we set the root nodes as the query variables, and the unsafe behaviours as evidence variables. The diagnostic inference results are shown in Figure 5.

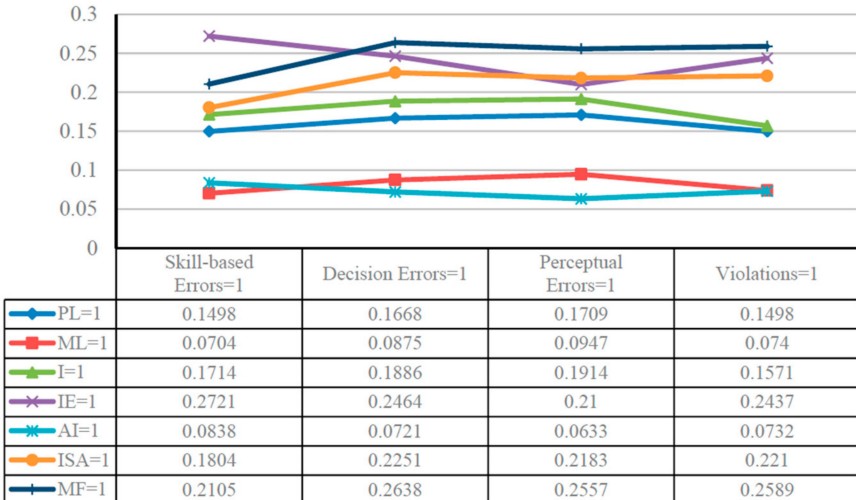

| | Skill-based Errors=1 | Decision Errors=1 | Perceptual Errors=1 | Violations=1 |
|---|---|---|---|---|
| PL=1 | 0.1498 | 0.1668 | 0.1709 | 0.1498 |
| ML=1 | 0.0704 | 0.0875 | 0.0947 | 0.074 |
| I=1 | 0.1714 | 0.1886 | 0.1914 | 0.1571 |
| IE=1 | 0.2721 | 0.2464 | 0.21 | 0.2437 |
| AI=1 | 0.0838 | 0.0721 | 0.0633 | 0.0732 |
| ISA=1 | 0.1804 | 0.2251 | 0.2183 | 0.221 |
| MF=1 | 0.2105 | 0.2638 | 0.2557 | 0.2589 |

**Figure 5.** The diagnostic inference results of the behaviour network.

Based on the aforementioned analysis and results, we can conclude that insufficient experience (IE) is the most likely cause of skill-based errors (SBE). Furthermore, mental limitations (ML) are the most important factor causing decision errors (DE), perceptual errors (PE), and violations (V).

Second, we set the unsafe behaviour as the query variable and the worker state as the evidence variable to provide a prediction inference (see Figure 6).

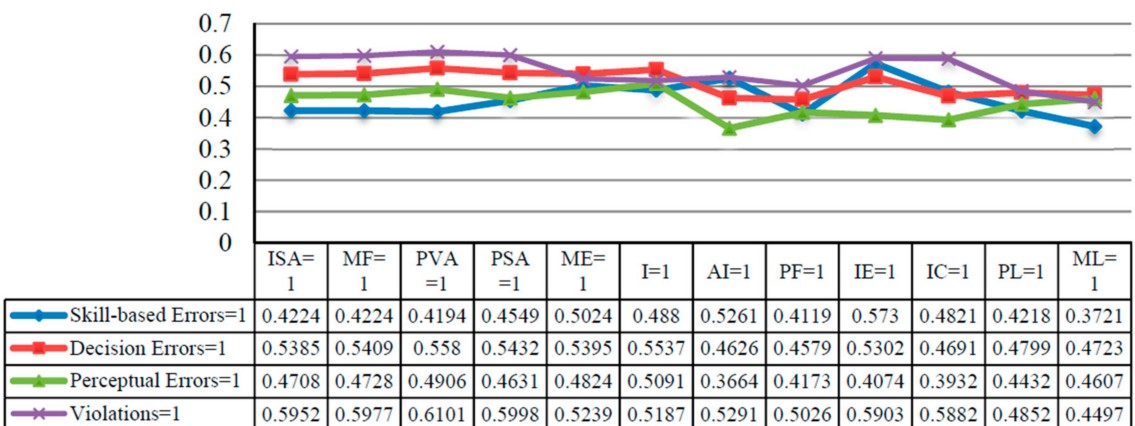

| | ISA=1 | MF=1 | PVA=1 | PSA=1 | ME=1 | I=1 | AI=1 | PF=1 | IE=1 | IC=1 | PL=1 | ML=1 |
|---|---|---|---|---|---|---|---|---|---|---|---|---|
| Skill-based Errors=1 | 0.4224 | 0.4224 | 0.4194 | 0.4549 | 0.5024 | 0.488 | 0.5261 | 0.4119 | 0.573 | 0.4821 | 0.4218 | 0.3721 |
| Decision Errors=1 | 0.5385 | 0.5409 | 0.558 | 0.5432 | 0.5395 | 0.5537 | 0.4626 | 0.4579 | 0.5302 | 0.4691 | 0.4799 | 0.4723 |
| Perceptual Errors=1 | 0.4708 | 0.4728 | 0.4906 | 0.4631 | 0.4824 | 0.5091 | 0.3664 | 0.4173 | 0.4074 | 0.3932 | 0.4432 | 0.4607 |
| Violations=1 | 0.5952 | 0.5977 | 0.6101 | 0.5998 | 0.5239 | 0.5187 | 0.5291 | 0.5026 | 0.5903 | 0.5882 | 0.4852 | 0.4497 |

**Figure 6.** The prediction inference for the worker state on unsafe behaviours.

We can conclude that:

(1)  Generally speaking, when a worker is in a poor state, the most vulnerable unsafe behaviour is violation, followed by decision-making errors. Furthermore, an inadequacy of safety awareness, mental fatigue, poor vigilance awareness, poor situational awareness, insufficient experience, and insufficient competencies are most likely to cause violations among operators. Hence, operators without enough training, or with no proficiency, are prone to violating regulations and procedures during work.

(2)  The effects of an inadequate safety awareness, mental fatigue, poor vigilance awareness, poor situational awareness, medical effect, illness, and insufficient experience on the four unsafe behaviours are greater, and the average impacts are 0.506725, 0.50845, 0.519525, 0.51525, 0.51205, 0.517375, and 0.525225, respectively. Therefore, coal mine enterprises should establish effective measures to curb the occurrence of these states, by for instance strengthening education and training to raise awareness of the safety of operators, implementing job rotation to alleviate the mental stresses endured by employees when working in the same position, perfecting leave management to prevent illnesses affecting operators' work, etc.

(3)  Insufficient experience is the most important factor affecting unsafe behaviour. Hence, coal mine enterprises should promote measures, such as improved wages and working environments, for operators, with a view to decreasing the staff turnover.

In particular, these worker states can be divided into four basic categories: adverse mental states (AMS), adverse physiological states (APS), fitness for duty (FD), and physical/mental limitations (PML). We can further investigate their influences on unsafe worker behaviours (see Table 4).

**Table 4.** The effects of AMS, APS, FD, and PML on unsafe worker behaviours.

| | Skill-Based Errors = 1 | Decision Errors = 1 | Perceptual Errors = 1 | Violations = 1 |
|---|---|---|---|---|
| AMS = 1 | 0.5631 | 0.4338 | 0.7181 | 0.6596 |
| APS = 1 | 0.5642 | 0.4404 | 0.5027 | 0.5763 |
| FD = 1 | 0.7008 | 0.7400 | 0.6665 | 0.6042 |
| PLM = 1 | 0.5211 | 0.4042 | 0.4725 | 0.5425 |

Fitness for duty (FD), such as insufficient competencies, insufficient experience, medication effect, and alcoholic intoxication, is the principal state that causes unsafe behaviours; in particular, the impact of fitness for duty on decision errors is the most significant. Hence, a business should implement strict job requirements preventing those who cannot perform from trying to at the cost of being a risk to others. Second, the impact of the four category states on violations is the biggest, followed by perceptual errors and skill-based errors.

## 7. Conclusions

Unsafe behaviours are commonly identified as important causal factors in coal mine accidents. Meanwhile, a recurring conclusion of accident investigations is that the worker state is an important contributory factor to unsafe behaviours. Hereby, it is interesting to analyse the impact of worker states on unsafe behaviours. We based this study on Bayesian networks in order to quantify the effects on coal mining accidents from a case study in China.

Based on accident investigation reports and expert opinions, we refined various worker states related to unsafe behaviours and used a Bayesian network to represent the cause-effect relationships between the states and unsafe behaviours. With the help of experts, we promoted a simple, graphical structure for the network. In particular, we proposed a verbal versus numerical fuzzy probability assessment method to elicit the conditional probability of the Bayesian network because of the insufficiency of data. Consequently, the junction tree algorithm was further used in the analysis. We showed that when a worker was in a poor state, the most vulnerable unsafe behaviour was violation, followed by decision-making errors. The conclusion is consistent with [18], which determined the weights of the accident-causing factors in China's coal mine accidents based on the HFACS model and AHP method. Furthermore, in this work, insufficient experience is the most significant contributory factor to unsafe behaviour. Meanwhile, in [23], safety awareness is ranked as the most influencing factor of unsafe behaviors of coal miners, followed by experience. Finally, poor fitness for duty is the principal state that causes unsafe behaviours. As shown, Bayesian networks provide a useful method to analyse the relationships between worker states and unsafe behaviours.

The limitation of this work is that the proposed "behavioural network" was built based on the input from a limited number of experts, due to the lack of a large amount of worker state data. Future research should monitor the state of workers in real-time. Indeed, wearable devices, such as Google Glasses and iWatches, will allow us to gather large data sets about workers' health, habits, etc. in real-time. Therefore, we can build a dynamic Bayesian network to evaluate the risk level of the unsafe behaviours of operators and propose some effective measures for reducing the risk to an acceptable level.

**Author Contributions:** Z.C. did the formal analysis and wrote the original draft; G.Q. revised the written draft and provided theoretical and technical guidance; J.Z. provided revision advice.

**Funding:** This research was supported by The Humanity and Social Science Youth foundation of Ministry of Education of China under Grants 15YJC630012.

**Conflicts of Interest:** The authors declare no conflict of interest.

# Appendix A

**Table A1.** The relationships between the variables of the behaviour network.

| | PL | ML | AI | I | IE | MF | ISA | AMS | APS | PLM | FD | ME | PF | PVA | PSA | IC | V | PE | DE | SBE |
|---|----|----|----|---|----|----|-----|-----|-----|-----|----|----|----|-----|-----|----|---|----|----|-----|
| PL | | | | | | | | ↑ | ↑ | → | ↑ | ↑ | ↑ | ↑ | ↑ | ↑ | | | | |
| ML | | | | | | | | ↑ | ↑ | → | ↑ | ↑ | ↑ | ↑ | ↑ | → | | | | |
| AI | | | | | | | | ↑ | ↑ | ↑ | → | ↑ | → | ↑ | ↑ | ↑ | | | | |
| I | | | | | | | | ↑ | → | ↑ | ↑ | → | → | ↑ | ↑ | ↑ | | | | |
| IE | | | | | | | | ↑ | ↑ | ↑ | → | ↑ | ↑ | ↑ | → | ↑ | | | | |
| MF | | | | | | | | → | ↑ | ↑ | ↑ | ↑ | ↑ | → | ↑ | ↑ | | | | |
| ISA | | | | | | | | → | ↑ | ↑ | ↑ | ↑ | ↑ | → | ↑ | ↑ | | | | |
| AMS | ↑ | ↑ | ↑ | ↑ | ↑ | ← | ← | | ↑ | ↑ | ↑ | ↑ | ↑ | ← | ← | ↑ | → | → | → | ↑ |
| APS | ↑ | ↑ | ↑ | ← | ↑ | ↑ | ↑ | ↑ | | ↑ | ↑ | ↑ | ← | ↑ | ↑ | ↑ | ↑ | → | → | ↑ |
| PML | ← | ← | ↑ | ↑ | ↑ | ↑ | ↑ | ↑ | ↑ | | ↑ | ↑ | ↑ | ↑ | ↑ | ↑ | ↑ | → | → | ↑ |
| FD | ↑ | ↑ | ← | ↑ | ← | ↑ | ↑ | ↑ | ↑ | ↑ | | ← | ↑ | ↑ | ↑ | ← | → | | → | → |
| ME | ↑ | ↑ | ↑ | ← | ↑ | ↑ | ↑ | | | | | | → | ↑ | ↑ | ↑ | | | | |
| PF | ↑ | ↑ | ← | ← | ↑ | ↑ | ↑ | | | | | ← | | → | ↑ | ↑ | | | | |
| PVA | ↑ | ↑ | ↑ | ↑ | ↑ | ← | ↑ | | | | | ↑ | ← | | → | ↑ | | | | |
| PSA | ↑ | ↑ | ↑ | ↑ | ← | ↑ | ↑ | | | | | ↑ | ↑ | ← | | ← | | | | |
| IC | ↑ | ← | ↑ | ↑ | ↑ | ↑ | ↑ | | | | | ↑ | ↑ | ↑ | → | | | | | |
| V | | | | | | | | ↑ | ↑ | ↑ | ← | | | | | | | | | |
| PE | | | | | | | | ← | ← | ← | ↑ | | | | | | | | | |
| DE | | | | | | | | ← | ← | ← | ← | | | | | | | | | |
| SBE | | | | | | | | ← | ↑ | ↑ | ← | | | | | | | | | |

*a ↑ b indicates that there is no direct causal relationship between *a* and *b*, *a* → b indicates that *a* directly cause *b*, *a* ← b indicates that *b* directly cause *a*. * The empty cells indicate that it was not necessary to confirm the relationship based on experts' knowledge because of the assumptions that there were no relationships between there pairs nodes.

**Table A2.** The behaviour network parameters.

| PL | ML | PML | Parameter | ISA | MF | PSA | PVA | AMS | Parameter |
|----|----|-----|-----------|-----|----|-----|-----|-----|-----------|
| 0 | 0 | 1 | 0.02 | 0 | 0 | 0 | 0 | 1 | 0.03 |
| 0 | 1 | 1 | 0.44 | 0 | 0 | 0 | 1 | 1 | 0.39 |
| 1 | 0 | 1 | 0.58 | 0 | 0 | 1 | 0 | 1 | 0.36 |
| 1 | 1 | 1 | 0.95 | 0 | 0 | 1 | 1 | 1 | 0.56 |
| **PF** | **I** | **APS** | **P** | 0 | 1 | 0 | 0 | 1 | 0.48 |
| 0 | 0 | 1 | 0.08 | 0 | 1 | 0 | 1 | 1 | 0.69 |
| 0 | 1 | 1 | 0.28 | 0 | 1 | 1 | 0 | 1 | 0.68 |
| 1 | 0 | 1 | 0.74 | 0 | 1 | 1 | 1 | 1 | 0.77 |
| 1 | 1 | 1 | 0.85 | 1 | 0 | 0 | 0 | 1 | 0.43 |
| **FD** | **AMS** | **V** | **P** | 1 | 0 | 0 | 1 | 1 | 0.62 |
| 0 | 0 | 1 | 0.08 | 1 | 0 | 1 | 0 | 1 | 0.59 |
| 0 | 1 | 1 | 0.64 | 1 | 0 | 1 | 1 | 1 | 0.74 |
| 1 | 0 | 1 | 0.53 | 1 | 1 | 0 | 0 | 1 | 0.72 |
| 1 | 1 | 1 | 0.83 | 1 | 1 | 0 | 1 | 1 | 0.85 |
| | **ML** | **IC** | **parameter** | 1 | 1 | 1 | 0 | 1 | 0.82 |
| | 0 | 1 | 0.69 | 1 | 1 | 1 | 1 | 1 | 0.94 |
| | 1 | 1 | 0.39 | **IE** | **IC** | **AI** | **ME** | **FD** | **parameter** |
| | **I** | **ME** | **parameter** | 0 | 0 | 0 | 0 | 1 | 0.01 |
| | 0 | 1 | 0.05 | 0 | 0 | 0 | 1 | 1 | 0.29 |
| | 1 | 1 | 0.73 | 0 | 0 | 1 | 0 | 1 | 0.26 |
| | **FD** | **SBE** | **parameter** | 0 | 0 | 1 | 1 | 1 | 0.45 |
| | 0 | 1 | 0.22 | 0 | 1 | 0 | 0 | 1 | 0.38 |
| | 1 | 1 | 0.74 | 0 | 1 | 0 | 1 | 1 | 0.62 |

**Table A2.** *Cont.*

| PL | | ML | PML | Parameter | ISA | MF | PSA | PVA | AMS | Parameter |
|---|---|---|---|---|---|---|---|---|---|---|
| IE | IC | PVA | PSA | parameter | 0 | 1 | 1 | 0 | 1 | 0.58 |
| 0 | 0 | 0 | 1 | 0.02 | 0 | 1 | 1 | 1 | 1 | 0.70 |
| 0 | 0 | 1 | 1 | 0.36 | 1 | 0 | 0 | 0 | 1 | 0.42 |
| 0 | 1 | 0 | 1 | 0.43 | 1 | 0 | 0 | 1 | 1 | 0.69 |
| 0 | 1 | 1 | 1 | 0.60 | 1 | 0 | 1 | 0 | 1 | 0.64 |
| 1 | 0 | 0 | 1 | 0.64 | 1 | 0 | 1 | 1 | 1 | 0.72 |
| 1 | 0 | 1 | 1 | 0.72 | 1 | 1 | 0 | 0 | 1 | 0.75 |
| 1 | 1 | 0 | 1 | 0.81 | 1 | 1 | 0 | 1 | 1 | 0.87 |
| 1 | 1 | 1 | 1 | 0.89 | 1 | 1 | 1 | 0 | 1 | 0.85 |
| MF | ISA | PF | PVA | parameter | 1 | 1 | 1 | 1 | 1 | 0.92 |
| | | | | | AMS | APS | PML | FD | DE | parameter |
| 0 | 0 | 0 | 1 | 0.06 | 0 | 0 | 0 | 0 | 1 | 0.01 |
| 0 | 0 | 1 | 1 | 0.4 | 0 | 0 | 0 | 1 | 1 | 0.32 |
| 0 | 1 | 0 | 1 | 0.43 | 0 | 0 | 1 | 0 | 1 | 0.34 |
| 0 | 1 | 1 | 1 | 0.74 | 0 | 0 | 1 | 1 | 1 | 0.42 |
| 1 | 0 | 0 | 1 | 0.45 | 0 | 1 | 0 | 0 | 1 | 0.41 |
| 1 | 0 | 1 | 1 | 0.78 | 0 | 1 | 0 | 1 | 1 | 0.51 |
| 1 | 1 | 0 | 1 | 0.83 | 0 | 1 | 1 | 0 | 1 | 0.58 |
| 1 | 1 | 1 | 1 | 0.92 | 0 | 1 | 1 | 1 | 1 | 0.68 |
| AI | I | ME | PF | parameter | 1 | 0 | 0 | 0 | 1 | 0.56 |
| 0 | 0 | 0 | 1 | 0.74 | 1 | 0 | 0 | 1 | 1 | 0.59 |
| 0 | 0 | 1 | 1 | 0.4 | 1 | 0 | 1 | 0 | 1 | 0.64 |
| 0 | 1 | 0 | 1 | 0.48 | 1 | 0 | 1 | 1 | 1 | 0.75 |
| 0 | 1 | 1 | 1 | 0.7 | 1 | 1 | 0 | 0 | 1 | 0.73 |
| 1 | 0 | 0 | 1 | 0.65 | 1 | 1 | 0 | 1 | 1 | 0.79 |
| 1 | 0 | 1 | 1 | 0.85 | 1 | 1 | 1 | 0 | 1 | 0.82 |
| 1 | 1 | 0 | 1 | 0.89 | 1 | 1 | 1 | 1 | 1 | 0.92 |
| 1 | 1 | 1 | 1 | 0.92 | | | | | | |

| PLM | APS | AMS | PE | parameter | ISA | parameter | | MF | parameter |
|---|---|---|---|---|---|---|---|---|---|
| 0 | 0 | 0 | 1 | 0.18 | 1 | 0.18 | | 1 | 0.21 |
| 0 | 0 | 1 | 1 | 0.31 | I | parameter | | PL | parameter |
| 0 | 1 | 0 | 1 | 0.39 | 1 | 0.15 | | 1 | 0.15 |
| 0 | 1 | 1 | 1 | 0.55 | ML | parameter | | IE | parameter |
| 1 | 0 | 0 | 1 | 0.45 | 1 | 0.08 | | 1 | 0.2 |
| 1 | 0 | 1 | 1 | 0.69 | AI | parameter | | | |
| 1 | 1 | 0 | 1 | 0.75 | 1 | 0.06 | | | |
| 1 | 1 | 1 | 1 | 0.78 | | | | | |

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
