# Peer review of "Study on the Relationship between Worker States and Unsafe Behaviours in Coal Mine Accidents Based on a Bayesian Networks Model"

_sustainability, doi:10.3390/su11185021_

Round 1

Reviewer 1 Report

The authors present a very interesting paper, focusing on an important work safety topic that needs to be dealt with more diligently.

The authors cite very up-to-date references, which is a sign they are, themselves, up to date on the matter.

The article is titled "Impact of worker status on unsafe behaviour in coal mine accidents". From the Abstract, one reads: "In this article, we seek to quantify the effects of operator working conditions on unsafe behaviours in coal mine accidents based on case studies drawn from Chinese practice.", although Introduction, it is written: "In this article, we seek to make quantify the impact operator conditions on unsafe behaviour in coal mine accidents based on a case study in China." – this incoherence needs to be corrected. This sentence also serves to show one of the errors that motivate the need to perform an English language correction (similar errors in lines 45, 46, 150 (“Notwithstanding this […]”). Also, “Operator conditions” seems to be used interchangeably with “worker status”, for the sake of congruence, opt for one of the terminologies of better define them apart.

The authors collect their variables through 163 accident investigation reports from the Fenxi Coal Mine Safety Bureau in Shanxi, and refine four direct and indirect worker statuses related to unsafe behaviour, but do not clarify how they perform this refinement. This is all the more important because these documents are not accessible, one has to take their word for it that they exist and are sound and eligible for doing science with, so more information should be provided. The same remark can be made regarding the survey of three experts from the Fenxi Coal Mine Safety Bureau, whereby the authors identified another eight variables for the behaviour network. Moreover, the description of such variables is not clear enough, for instance, why is not the “misinterpretation of information” a Perceptual Error? It is not clear where the thresholds lie. Likewise, “Lack of Vigilance”, unlike other variables, is described by the consequence, and not the cause. It is not clear why the worker “Cannot detect the possible occurrence of accidents or signs of danger in time”.

The use of verb tenses in incoherent, shifting between future and present throughout the presentation of the work that was performed. It could be a good idea to use a past tense, since the work has already been done.

The authors repeat the caveat thought too many times throughout the paper regarding the lack of needed information to perform “an up-to-date data collection large and rich enough to allow for reliable assessment of the conditional probabilities based on the aforementioned methods”, regarding worker statuses.

Reviewer 2 Report

Thanks for presenting an interesting approach in your paper. I believe that if you revise the paper based on my comments and you pay more attention to explanations as well as your implied assertiveness, the study would be publishable.

Round 2

Reviewer 1 Report

The changes to the paper were significant and I believe that the authors have provided all the information they were able to provide. The flow of the information is much clearer, as is the methodology applied and the results obtained.